# Design and deployment of a fast neural network for measuring the properties of muons originating from displaced vertices in the CMS Endcap Muon Track Finder

**Efe Yigitbasi[1]★, Darin Acosta[1], Osvaldo Miguel Colin[1], Aleksei Greshilov[1], Sergo Jindariani[2], Patrick Kelling[1], Jacobo Konigsberg[3], Jia Fu Low[3], Alexander Madorsky[3][a] on behalf of CMS Collaboration**

**1** Rice University
**2** Fermi National Accelerator Laboratory
**3** University of Florida

★ efe.yigitbasi@cern.ch

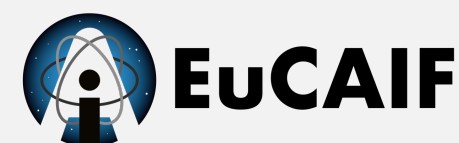

*The 2nd European AI for Fundamental Physics Conference (EuCAIFCon2025) Cagliari, Sardinia, 16-20 June 2025*

## Abstract

**We report on the development, implementation, and performance of a fast neural network used to measure the transverse momentum in the CMS Level-1 Endcap Muon Track Finder. The network aims to improve the triggering efficiency of muons produced in the decays of long-lived particles (LLPs). We implemented it in firmware for a Xilinx Virtex-7 FPGA and deployed it during the LHC Run 3 data-taking in 2023. The new displaced muon triggers that use this algorithm broaden the phase space accessible to the CMS experiment for searches that look for evidence of LLPs that decay into muons.**

## 1 Introduction

Long-lived particles (LLPs) that have large mean lifetimes are predicted by many extensions to the Standard Model (SM). These LLPs, when produced in collisions at the Large Hadron Collider (LHC) at CERN, can be observed with the CMS detector [1] through their decays to SM particles at macroscopic distances from the interaction point (IP). The resulting signature is often different from that of promptly decaying particles, and requires dedicated triggering techniques and algorithms. In this paper we describe the design, deployment, and performance of a fast neural network (NN) based algorithm implemented in the CMS Level-1 Trigger (L1T) [2,3] Endcap Muon Track Finder (EMTF), performing transverse momentum ($p_T$) regression targeting muons originating from decays of LLPs.

---

[a]now at Boston University

The CMS L1T is the first level of the two-level CMS trigger system, which is designed to select interesting physics processes out of 40 MHz of LHC collisions. The L1T is implemented on custom designed electronics, and it selects interesting collision events with a latency of about 4 $\mu$s using the reduced readout from the detectors. The EMTF is one of the three muon track finders in the CMS L1T that builds muon tracks and measures their kinematic properties and electric charge within a $\sim 500$ ns total latency budget. The CMS endcaps are a particularly challenging environment for triggering due to factors such as non-uniform magnetic field in the endcaps, the different detector technologies with different spatial and timing resolutions, and large collision backgrounds that increase with increasing $\eta$ and can lead to a non-linear dependence of trigger rate with respect to number of simultaneous interactions in a collision event. These challenges create an ideal problem for ML based solutions, which have been used in the EMTF system since Run 2 of the LHC in the form of a Boosted Decision Tree (BDT) based $p_T$ assignment algorithm that was optimized for prompt muons. In Run 3 of the LHC, we introduced a new NN-based algorithm to measure the $p_T$ and transverse impact parameter ($d_0$) of muon tracks originating from displaced vertices.

The following sections will describe the design, hardware implementation, deployment for operations, and the expected performance of the EMTF displaced NN algorithm.

## 2 Design and Hardware Implementation

Although the BDT-based $p_T$ assignment algorithm is very capable for measuring the $p_T$ of prompt muons, its accuracy falls sharply with increasing muon $d_0$ after $\sim 20$ cm. This result is due to the training samples used for the BDT, which contain only prompt muons, and the BDT learning to extrapolate the muon track to the IP while estimating the muon $p_T$ essentially. In order to increase the trigger acceptance to highly displaced muons, a new ML algorithm that is trained with samples containing displaced muons is required. Additionally it is preferred to have an algorithm that does not modify the BDT-based method so as not to interfere with the prompt muon triggering. Since there is also no spare latency in the EMTF budget, the resulting decision is to implement an ML-based algorithm that runs parallel to the BDT-based $p_T$ assignment within the $\sim 100$ ns latency used by the BDT stage.

The Run 3 EMTF algorithm is implemented in Virtex 7 Field Programmable Gate Arrays (FPGAs). The resource usage of the EMTF FPGA before Run 3 was as follows:

- 74% of FPGA Look-Up Tables (LUTs)

- 2% of Digital Signal Processors (DSPs)

- 76% Block RAMs (BRAMs)

- 25% Flip Flops

This implies that while implementing the new ML algorithm there is virtually no constraint for using DSP resources, while the LUT and BRAM resources are already significantly constrained. This makes an NN-based solution ideal for Run 3 EMTF since NNs can be implemented in FPGAs by using mostly DSP resources.

Therefore an NN-based solution is chosen to be implemented in Run 3 EMTF system that targets measurement of $p_T$ and $d_0$ of displaced muons. Two separate NNs are developed in software to target $p_T$ and $d_0$ separately. Each of the NN models have three dense layers, and equally split half of the total nodes of the complete model as seen in Fig. 1. The two models are then stitched together to form a single NN model before being implemented into the FPGA. The models are trained on simulations of LLPs that decay to muons made with standard CMS

software emulators. The trainings are done separately using $\sim 1.5$ M events, each containing 4 muons, both minimizing similar log-cosh loss functions over 300 epochs. The 29 input features are calculated for each muon track, and include $\theta$ of the track measured from the beam axis, differences of $\phi$ and $\theta$ angles and their signs between each stub used to build the track, and a parameter that measures the bending recorded by each stub.

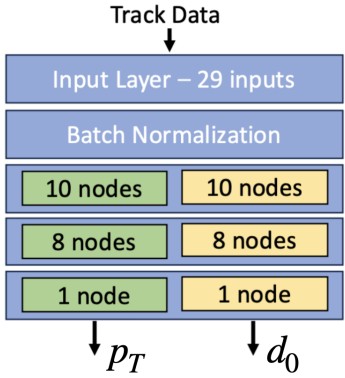

Figure 1: The overall design of the complete displaced NN model. The complete model is obtained by stitching together two identical models with three dense layers and total of 19 nodes with one model targeting $p_T$ and the other targeting $d_0$.

After the training, the complete model is quantized to fixed point precision and implemented into FPGAs using hls4ml [4]. In order to fit the NN model within the latency constraints, a wrapper around the NN model with fixed latency that also handles input/output conversions was implemented for the final design. The quantization of the bit-widths for weights and activations are chosen for optimal performance, since the DSP resource savings resulting from reducing the bit-widths further was irrelevant due to the available resources in the FPGAs. Accumulation bit-widths were optimized to reduce the LUT usage without impacting the performance significantly.

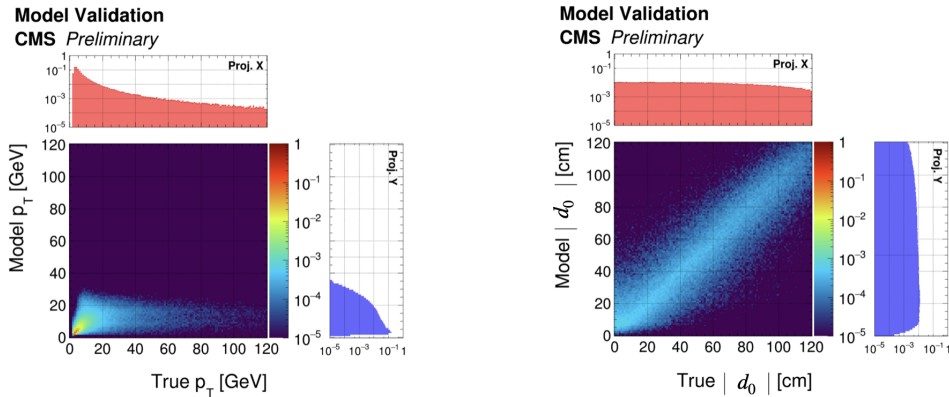

Figure 2: The validation plots comparing measured and true $p_T$ (left) and $d_0$ (right).

Final implementation of the NN model fits into a latency of 83 ns. After the NN implementation the LUT usage in the FPGA is 76% (from 74%) and the DSP usage is 14% (from 2%). The validation of the model shows good agreement between $p_T$ and $d_0$ measured by the NN model and the true values as shown in Fig. 2. The underestimation of $p_T$ at higher true $p_T$ is not crucial since the current displaced muon trigger designs target 5-10 GeV range which has good accuracy.

## 3   Deployment for Operations and Expected Performance

The displaced NN was deployed for CMS operations on 26th June 2023, making it the first NN running online in CMS L1T FPGAs during data taking. The commissioning of the NN was performed during the 2023 data taking year using monitoring triggers such as the one seen in Fig. 3, which shows that the deployed NN works with a rate vs pile-up (PU) profile as expected from simulations. Fig. 3 also shows the expected performance of the displaced NN (solid stars) and prompt BDT algorithms (hollow squares) on a simulated sample of displaced muons produced with an earlier design of the NN model from 2022 [5][b]. The different colors show different $\eta$ regions: $1.2 < |\eta| < 1.6$ (black), $1.6 < |\eta| < 2.1$ (red), and $2.1 < |\eta| < 2.5$ (blue). The first set of triggers making use of the improvements from the NN based $p_T$ and $d_0$ assignment were deployed for data taking in 2024. These new triggers increase the CMS trigger acceptance to highly displaced muons ($d_{xy} > 20$ cm), which substantially improve signal sensitivities to models with LLPs that decay to muons in the final state.

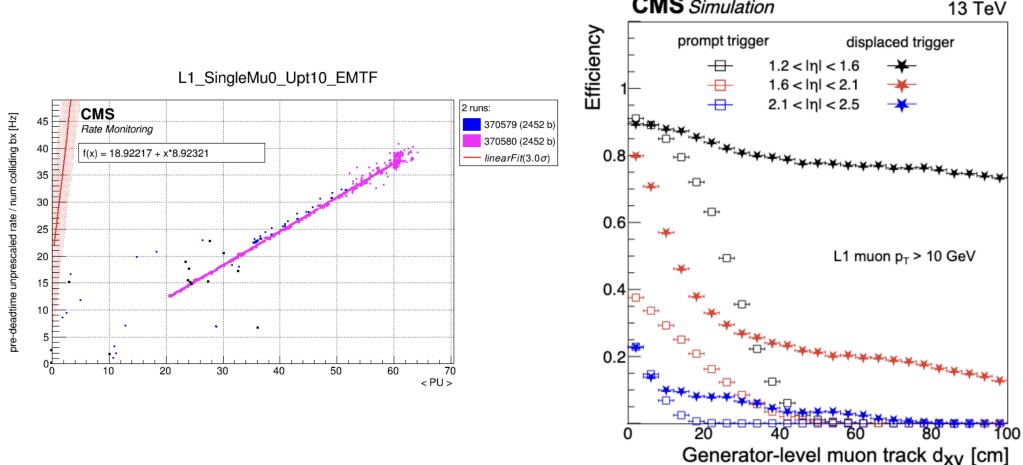

Figure 3: The rate vs PU plot for one of the monitoring triggers recorded in LHC proton-proton collisions from July 2023 (left) which requires one muon in EMTF acceptance with $p_T$ assigned by the displaced NN greater than 10 GeV and no requirement on the $p_T$ assigned by the prompt BDT algorithm. The EMTF trigger efficiencies for prompt and displaced-muon algorithms for L1 $p_T > 10$ GeV with respect to muon track $d_{xy}$ obtained using a displaced-muon simulation sample (right) [5].

## 4   Conclusion

The CMS experiment aims to broaden the accessible phase space for searches that look for evidence of LLPs in the Run 3 of the LHC. A new fast NN-based algorithm to improve the efficiency of triggering on muons produced in the decays of LLPs was developed for the EMTF system, which was deployed for data-taking in 2023. The new NN-based algorithm offers significant improvements to triggering efficiency of muons with $d_0 > 20$ cm compared to the prompt BDT-based method. The new triggers based on this algorithm will increase the physics reach of the CMS experiment for the remainder of the Run 3 of the LHC.

---

[b]A more recent version of this plot was published in CMS-PAS-EXO-23-016

## 5 Acknowledgments

This paper is based upon work supported in part by the U.S. Department of Energy, DOE Grants: DE-SC0023351 and DE-SC0010103

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
