# Peer review of "Design and deployment of a fast neural network for measuring the properties of muons originating from displaced vertices in the CMS Endcap Muon Track Finder"

_SciPost Physics Proceedings_

## Round 1 · Referee Report · Anonymous (Referee 1) · 2025-10-30

Strengths

1-It is great to see a neural network running on an FPGA within an LHC experiment.
2-The motivation, goal, and constraints of the project are clearly stated and well-explained.
3-The project achieved significant improvement compared to the baseline method

Weaknesses

1-Very little information is provided on the network itself: what are the inputs, how was it trained, what loss function is being minimised?
2-The result could be commented on a bit more
3-The potential impact of the new algorithm on physics performance is not mentioned (understandable for a 4-page proceeding)

Report

I think this paper is well written and very interesting. I recommend it for publication. I have a few comments on some additional information that could be included in the paper, which I have added below.

Requested changes

1-Would it be possible to provide a bit more information on the network itself? In decreasing order of importance, it would be helpful to see: what are the input features ('track features’ is quite vague), how it was trained (e.g., number of events, epochs), and what loss was considered.
2-It would be interesting to comment a bit more on the result, as it is currently only mentioned to be better.
3-Concerning Figure 3, I am not entirely sure I fully understand the message conveyed by the left plot and how it relates to the rest of the paper. I also think the description of the right plot could be made significantly shorter, and the information it contains could be merged into Section 3.

Recommendation

Publish (meets expectations and criteria for this Journal)

  • validity: top
  • significance: high
  • originality: good
  • clarity: high
  • formatting: good
  • grammar: good

Author:  Efe Yiğitbaşı  on 2025-11-21  [id 6061]

(in reply to Report 1 on 2025-10-30)

Thank you for the report. I have implemented some changes following your requests.

  1. I added a brief description of the NN training, summarizing input features and how the NN was trained.
  2. I added a brief comment on the resulting improvement. I hope this is enough, since a longer description of expected improvements on LLP signals would not realistically fit to this proceedings.
  3. I added a description on how the left figure shows that the deployed NN works in online operations as expected from simulations. I also moved some of the text from caption to the main body.

The new version is already submitted as a resubmission.

---

## Editorial Decision

in_voting